# Association of Low Serum 25OHD Levels with Abnormal Bone Microarchitecture in Well-Differentiated Thyroid Cancer

**DOI:** 10.3390/medsci8040049

**Published:** 2020-12-01

**Authors:** Federico Hawkins Carranza, Sonsoles Guadalix Iglesias, María Luisa De Mingo Dominguez, Gonzalo Allo Miguel, Cristina Martín-Arriscado Arroba, Begoña López Alvares, Guillermo Martínez Diaz-Guerra

**Affiliations:** 1Research Institute i+12, University Hospital 12 de Octubre, Faculty of Medicine, Complutense University, 28041 Madrid, Spain; 2Endocrinology Service, University Hospital 12 de Octubre, 28041 Madrid, Spain; sonsoles.guadalix@salud.madrid.org (S.G.I.); gonzalo.allo@salud.madrid.org (G.A.M.); guillermo.martinez@salud.madrid.org (G.M.D.-G.); 3La Luz Hospital, 28003 Madrid, Spain; marisadmd@hotmail.com; 4Epidemiology Unit, Research Institute i+12, University Hospital 12 de Octubre, 28041 Madrid, Spain; cmartinarriscado.imas12@h12o.es; 5Centro de Salud Goya, 28009 Madrid, Spain; begolopezal@gmail.com

**Keywords:** 25 hydroxyvitamin D, differentiated thyroid cancer, bone mineral density, trabecular bone structure, TSH suppression therapy

## Abstract

The association of low levels of 25 hydroxyvitamin D (25OHD) with papillary thyroid cancer (PTC) is being studied, as to whether it is a risk factor or as a coincidental one. This study aimed to evaluate serum levels of deficiency, insufficiency, and sufficiency of 25OHD in PTC and its relationship with the trabecular bone score (TBS) and bone mineral density (BMD). This study includes 134 postmenopausal women with PTC, followed for 10 years. BMD was measured with DXA Hologic QDR 4500, and TBS with Med-Imaps iNsight2.0 Software. Mean serum 25OHD was 23.09 ± 7.9 ng/mL and deficiency, insufficiency, and sufficiency levels were 15.64 ± 2.9, 25.27 ± 2.7, and 34.7 ng/mL, respectively. Parathyroid hormone (PTH) and bone alkaline phosphatase (BAP) were higher in deficiency (57.65 ± 22.6 ng/mL; 29.5 ± 14 U/L) and in insufficiency (45.88 ± 19.8 ng/mL; 23.47 ± 8.8 U/L) compared with sufficiency of 25OHD (47.13 ± 16 and 22.14± 9.7 ng/mL) (*p* = 0.062 and *p* = 0.0440, respectively). TBS was lower in patients with 25OHD < 20 ng/mL (1.24 ± 0.13) compared with between 20–29 (1.27 ± 0.13, *p* < 0.05) and 30 ng/mL (1.31 ± 0.11, *p* < 0.01). We found low TBS in patients with PTC and long-term follow-up associated with low serum 25OHD levels, not associated with cancer stage, or accumulative iodine radioactive dose. Low 25OHD associated with deleterious bone quality in patients with PTC should be restored for the prevention of fractures.

## 1. Introduction

Thyroid cancer represents approximately 1% of all cancers and more than 90% of all endocrine malignancies [1]. The prevalence of this tumor has increased worldwide. However, it has been discussed whether this rise reflects earlier detection of subclinical disease (with ultrasound and fine needle cytology) or a real increase in the rate of this type of cancer. It has been proposed that vitamin D deficiency could contribute to or be a potential risk factor for thyroid cancer [2,3]. Some studies have not found an association between vitamin D status and thyroid cancer [4,5,6]. However, a retrospective study has shown an increased rate of thyroid cancer in patients with insufficient serum 25OHD levels compared with patients with 25-hydroxyvitamin D (25OHD) sufficiency (37.5 and 75% ng/mL, respectively), suggesting that vitamin D may be a modifiable risk factor for thyroid cancer [7]. Also, decreased levels of serum 25OHD in 344 patients with PTC have been reported, compared with 116 healthy controls [8]. 

Two recent meta-analyses have reported an association between vitamin D deficiency and the risk of thyroid cancer. Hu et al. reported that plasma 25OHD concentration was inversely associated with thyroid cancer risk. Compared with non-deficient patients, the OR was 1.42 (95% confidence interval (CI): 1.17–1.73) in the vitamin D-deficient group [9]. In the second meta-analysis, Zhao et al. reported that lower serum 25OHD increased the risk of thyroid cancer by 30%, compared with non-deficient individuals [10]. 

Up to now, the ideal levels of circulation of 25OHD for optimal bone status are under discussion. The Institute of Medicine (IOM) has concluded that vitamin D deficiency should be considered with serum 25OHD below 20 ng/mL [11]. The Endocrine Society 2011 guidelines have proposed serum levels >30 ng/mL as the minimum level to maximize bone health (sufficiency), while levels between 20 to 29 ng/mL, are categorized as insufficiency [12]. Although bone mineral density (BMD) has been associated with bone loss in postmenopausal women with thyroid cancer after long-term TSH suppression therapy, up to now, there have been no data available for bone quality in patients with thyroid cancer regarding the status level of 25OHD [13]. This study aimed to evaluate the association between the level of serum 25OHD—deficiency, insufficiency, and sufficiency—in thyroidectomized patients due to PTC with long-term TSH suppression therapy, and its relationship with the trabecular bone score (TBS) and BMD.

## 2. Material and Methods

### 2.1. Design and Participants

In this single-center cross-sectional study, we enrolled postmenopausal Caucasian patients with a previous diagnosis of papillary thyroid cancer, treated with total thyroidectomy and ^131-^I ablation, who started levothyroxine (LT4) therapy 1–3 months after surgery, to achieve suppression of TSH. Exclusion criteria: (1) presence of post-surgical hypoparathyroidism; (2) treatment with estrogen replacement therapy, vitamin D, or other medications that might affect bone metabolism; (3) subjects who had any prior cancer history, thyroid autoimmune disease, daily vitamin supplementation, or a disease that would affect serum vitamin D levels were also excluded. The TNM staging system was used to define the low-risk cancer stage [14]. Patients had been before periodically controlled and the LT4 dose was adjusted to maintain serum TSH within the reference range (0.5–2.0 mU/L) in our center. Informed consent was obtained from all participants following the Declaration of Helsinki guidelines. The Ethics Committee of the University Hospital 12 de Octubre approved the study protocol.

### 2.2. Measurement and Biosamples

Fasting serum samples were obtained between 8 and 9 AM in the summer–autumn months and were immediately frozen at −70 °C until the assays were performed. Serum albumin-corrected calcium, phosphate, and creatinine were measured by automated standard laboratory methods (Modular P800 Chemistry Analyzer; Roche Diagnostics, Basil, Switzerland) and 24-h urinary calcium by colorimetric method. Serum 25–hydroxyvitamin D3 (25OHD) levels were measured by enzyme immunoassay (automated IDS EIA, kit, Roche Diagnostics, Basil, Switzerland). Endocrine Society 2011 guidelines categorization values have been used in this study: deficiency < 20 ng/mL; insufficiency ≥ 20 to 29 ng/mL; and sufficiency > 30 ng/mL (12).

Serum TSH was measured by chemiluminescence (Architect TSH reagent; Abbot Laboratories, San Francisco, CA, USA), free thyroxine (fT4) by electrochemiluminescence (Elecsys T4, Roche Diagnostics, Basil, Switzerland) (functional sensitivity < 0.01 µg/mL). Serum intact PTH (normal range: 7–57 pg/mL) was determined using chemiluminescent immunoassays with an Immulite 2000 (Siemens Healthcare, Erlangen, Germany).

### 2.3. Bone Mineral Density and Trabecular Bone Scores Measurements

BMD was assessed by DXA, (densitometer QDR 4500, Hologic, Waltham, MA, USA) at the lumbar spine (LS), femoral neck (FN), total hip (TH), and distal third of the radius (1/3DR), with a precision error ≤ 1.5%. BMD values are expressed as absolute values (g/cm2) and as standard deviations (SD) from the expected adult BMD for the T scores. According to the WHO criteria, patients were classified as osteoporotic (T score ≤ 2.5), osteopenic (T score −1 ≥ and >−2.5), and normal (T score > −1) [15]. Reference data corresponding to the Spanish population was obtained from a multicenter study with 2442 healthy subjects, aged 20–80 years [16]. The same equipment was used in all the studied patients.

TBS measurements were performed applying the TBS iNsight2.0 software (Med-Imaps, Geneva, Switzerland) to the LS DXA exams. Lumbar TBS was calculated as the mean value of individual measurements for vertebrae L1-L4. TBS measurements were performed applying the TBS iNsight2.0 software (Med-Imaps, Swiss) to the LS DXA exams. TBS was evaluated by determining the variogram of the trabecular bone projected image, calculated as the sum of the squared gray-level differences between pixels at a specific distance and angle. Lumbar TBS was calculated as the mean value of individual measurements for vertebrae L1–L4 [17].

Healthy subjects’ TBS values were obtained from a multicenter study developed by the Spanish Society of Mineral Metabolism Investigation (SEIOMM) in 8464 healthy volunteers (82% women). TBS categorization results in postmenopausal women were: ≥1.307 is considered normal; TBS between 1.307 and 1.215 is consistent with partially degraded microarchitecture; and TBS ≤ 1.215 with degraded microarchitecture (Del Rio L. et al. unpublished results). These values are in agreement with previously published TBS values [18]. The coefficient of variation of TBS, calculated from 3 repeated measurements in 15 women, was 0.8%.

### 2.4. Statistics

Quantitative variables are expressed as mean and standard deviations, or as medians with interquartile range. The normality of the data was confirmed using the Kolmogorov–Smirnov test. Qualitative variables were described using absolute and relative percentages. Contingency tables and the Chi-square or Fischer tests were used to compare categorical parameters. The non-parametric Wilcoxon test or the Kruskal–Wallis test were used for the cross-sectional study and the Student’s *t*-test for the longitudinal study. The Pearson test was performed to evaluate the correlations between serum 25OHD, clinical parameters, DXA, and TBS. Multiple linear regression was performed to evaluate the dependence and influence between variables. Adjusted covariates included menopausal status and body weight. The Bonferroni correction was used in the analysis of variance (ANOVA) tests. Fisher’s exact test was used to examine the significance of the association (contingency) between serum 25OHD and TNM stages.

The prediction model was performed using binary logistic regression, with the backward method, introducing a dependent variable having vitamin D greater than 30 ng/mL vs. having it between 20 and 30 ng/mL; and having vitamin D greater than 30 vs. having less than 20 and as independent variables those that obtained statistical significance in the bivariate analysis or could have a clinically plausible implication. Using the Wald index method, we study the permanence or exclusion of the variables. The results of the model are presented in the form of an odds ratio (95% confidence interval (CI)). A level of α = 0.05 was considered significant in all statistical procedures. All data were analyzed using SAS statistical software version 9.3 (SAS Institute, Cary, NC, USA).

## 3. Results

Demographic, biochemistry, and densitometry parameters are shown in Table 1. The study included 134 women with thyroid cancer and TSH suppression therapy with LT4; age 64.07 ± 10.8 years (after menopause: 12.98 ± 10.7); years of follow-up: 12.25 + 6.19 years. Serum-corrected calcium, creatinine, alkaline phosphatase, and 24-h urine calcium were in the normal range. Mean serum 25OHD level was in the range of insufficiency (23.09 ± 7.97 ng/mL). During the follow-up, 20 patients had 131-I treatment: 7 in the group with serum 25OHD below 20 ng/mL, 6 in the group with >30 ng/mL, and another 6 with serum levels >30 ng/mL, without intergroup significant differences in the total doses. Table 2 shows the distribution of serum 25OHD levels, according to the categorization of deficiency, insufficiency, and sufficiency. Serum PTH and bone alkaline phosphatase levels were higher in patients with 25OHD deficiency (57.65 ± 22.68 g/mL and 29.50 ± 14.04, U/L respectively), compared with patients with insufficiency (45.88 ± 19.8 pg/mL and 23.47 ± 8.8 U/L, *p* < 0.05 and *p* < 0.05 respectively) and sufficiency (47.13 ± 16, *p* = 0.062 and 22.14 ± 9.7, *p* < 0.05 respectively).

There were no differences in serum calcium, phosphate, creatinine, and 24-h urine calcium, between patients in the groups of vitamin D deficiency, insufficiency, and sufficiency. There were no significant differences in serum 25OHD levels according to BMI. Patients with BMI < 24.9 showed serum 25OHD levels of 26.32 ± 8.07 ng/mL, overweight patients (BMI: 25–29.9) of 27.73 ± 8.59 ng/mL, and obese patients (BMI > 30) of 21.71 ± 6.38 ng/mL. There were no differences in serum 25OHD levels, regarding smoking habit (smokers: 22.35 ± 8.0 ng/mL; non-smokers: 23.19 ± 7.8; *p* = 0.6330). Mean daily calcium ingestion was similar between the three groups studied (Table 2).

Regarding bone mineral density, patients with serum 25OHD deficiency showed significantly higher values in lumbar (L-) BMD, femoral neck (FN-) BMD, and total hip (TH-) BMD sites than patients with 25OHD insufficiency. At the radio site, we found significant differences in the 1/3DR-z score, TR-z score, and total-z score (Table 2a,b). Patients with sufficient vitamin D levels had higher values at lumbar and radius sites, compared with patients with 25OHD insufficiency. Patients with serum 25 OHD >30 ng/mL (*n* = 28) had significantly higher L-T score, FN-T score, and TH-T score than patients with levels <30 ng/mL (*n* = 106) (*p* < 0.005; *p* < 0.01, and *p* < 0.001 respectively). Also, there were more patients with serum 25OHDD < 30 ng/mL and osteoporosis than patients with >30 ng/mL (28% vs. 12.6%, respectively. Densitometry values according to WHO classification are represented in Table 3. The DXA study confirms the presence of osteoporosis and osteopenia in only 14.1% of patients with serum 25OHD > 30 ng/mL, while osteoporosis/osteopenia was reported in 66.4% of patients with levels of 25OHD > 30 ng/mL. There were no significant differences in the distribution of thyroid cancers according to the stage and levels of 25OHD (Figure 1).

TBS was significantly lower in patients with serum 25OHD < 20 ng/mL (1.24 ± 0.13) compared with patients with 25OHD levels between 20–29 ng/mL (1.27 ± 0.13; *p* < 0.05) and >30 ng/mL (1.31 ± 0.11; *p* < 0.01). TNM stage was not significantly related to serum 25OHD levels (Fisher’s test, *p* = 0.457).

Correlation analysis. There was no significant correlation between serum 25OHD and age (r = 0.058, *p* = 0.520). A significant negative correlation was found between 25OHD and BMI (r = −0.20, *p* < 0.05). Moreover, there was a negative correlation between serum fT4 levels and 25OHD (r = −0.21, *p* = 0.01). Serum 25OHD levels were significantly correlated with serum PTH levels (r = 0.18, *p* < 0.05) but not with serum TSH values (r = 0.16, *p* = 0.0620). Neither TBS (r = 0.12, *p* = 0.2038) nor L-BMD (r = −0.16, *p* = 0.0614) were correlated with serum 25OHD. There was a significant correlation between serum 25OHD levels and TH BMD (r = −0.19, *p* < 0.05), lumbar, and TH T-score (r = −0.19, *p* < 0.05 and r = −0.22, *p* = 0.01, respectively). Patient-accumulated radioactive iodine doses or histologic cancer stage showed no correlation with serum 25OHD levels (r = −0.03, *p* = 0.7500, and r = 0.0580, *p* = 0.5200, respectively).

Linear regression analysis. We performed a multivariable linear regression analysis with serum 25OHD as a dependent variable. TBS (b = 11.033, *p* < 0.001), duration of follow-up (b = −0.311, *p* < 0.001), and age (b = 0.076, *p* = 0.0143) were the most predictive values. We adjusted for BMD to show that the association of TBS and these parameters were independent.

Logistic regression analysis. 25OHD > 30 was used as the dependent variable. We found a protective Odds ratio for patients with serum 25OHD between 20–29 ng/mL of 0.030 (IC 0.002–0.421) for serum fT4 levels and 0.421 (IC 0.212–0.837) for TH z-score. For patients with serum 25OHD < 20 mg/mL, an odds ratio of 0.361 (IC 0.59–0.816) for TH z-score was also found in the analysis.

## 4. Discussion

In this study, we have found serum 25OHD levels of deficiency in 43.3%, and insufficiency in 35.8% of our thyroidectomized patients due to thyroid cancer, with long-term therapy with LT4 for an average of 10 years. This is the first study to demonstrate in these patients a deleterious trabecular bone status, associated with low levels of 25OHD.

The extra-skeletal actions of vitamin D have been evaluated before, and up to now, it has been shown to diminish tumor growth, with effects in cell-cycle progression, apoptosis, differentiation, and angiogenesis [19]. Also, vitamin D receptor (VDR) expression has been demonstrated in normal thyroid tissue and papillary thyroid cancer [20,21]. Therefore, vitamin D binding to the VDR at the thyroid gland could activate protection against carcinogenesis. The possible role of vitamin D deficiency in tumorigenesis is of main importance [19].

It has been proposed that vitamin D deficiency could contribute to or be a potential risk factor for thyroid cancer [4,22]. Observational and clinical studies have suggested the possible association between vitamin D deficiency and some types of cancers, with few studies referring to thyroid cancers. Roskies et al. were the first to report the association between the deficit of serum 25OHD and thyroid cancer in human subjects. They showed that 25OHD deficiency in patients with thyroid cancers is associated with a twofold risk of having a malignant thyroid nodule compared to patients with adequate 25OHD levels [7]. Heidari et al. reported an odds ratio eightfold higher for papillary thyroid cancer patients with levels below 20 ng/mL compared with patients higher to 20 ng/mL of serum 25OHD [3]. Other groups have found negative results. Stepien et al. reported 50 untreated thyroid cancer patients, with lower plasma concentrations of 1,25 OHD, respect healthy controls, without significant differences in plasma 25OHD concentrations [8].

Laney et al. reported no significant differences in serum 25OHD levels, between patients with 33 nodules and 57 thyroid cancers, considering the different histologic types of thyroid cancer [4]. In another study, female patients with thyroid cancer had lower preoperative serum 25OHD levels that were associated with poor clinicopathological features, although these significant differences were only found in the second and fourth quartiles for 25OHD, without evidence of linear relationship [23]. On the other hand, although vitamin D laboratory values were not predictive of thyroid cancer stage, documented vitamin D deficiency did show an association with thyroid cancer stage and with more advanced stage malignancy [24]. In our cohort, serum 25OHD levels in the whole and studied groups were not predictive of thyroid cancer stage. Bone alkaline phosphatase was higher in patients with vitamin D deficiency (29.50 ± 14) than in patients with insufficiency (23.47 ± 8.1) and sufficiency (22.14 ± 9.7) (*p* = 0.0233 and 0.0440, respectively); probably due to the increased PTH in the former group. We hypothesized that lower serum 25OHD (and the subsequent PTH increase) might explain the reduction in TBS due to the increase in bone remodeling.

It is also, necessary to consider that vitamin D deficiency is widely prevalent around the world. Our data of 66% of the cohort with a prevalence of serum <30 ng/mL of 25OHD, is lower with t reported data of 83% and 89% in two national series of Spain [25,26]. Therefore, low vitamin D is associated with postmenopausal women, whether they have thyroid cancer or not. The fact that postmenopausal women with thyroid cancer and thyroid hormone replacement, can have more loss of bone, makes this population a candidate to consider in TBS studies.

In our series, patients with thyroid cancer and with levels <20 ng/mL serum levels of 25OHD (1.24 ± 0.13) in the range of degraded TBS, had significantly lower values of TBS compared with patients with levels >30 ng/mL (1.31 ± 0.11, *p* = 0.01), in the normal range, and compared with patients with values 20–29 ng/mL, with partially degraded TBS (1.27 ± 0.13, *p* ≤ 0.05) after a mean 10 years of follow-up postsurgery. Recently, it has been reported in 1348 healthy postmenopausal women that TBS values were lower in patients with deficit and insufficiency of 25OHD, compared with patients with sufficient vitamin D levels measured at year five of follow-up. They found no significant differences in lumbar BMD between the three groups of vitamin D levels [27]. However, compared with women with levels of 25OHD below 20 ng/mL, those with higher levels had a significantly lower risk for hip fracture. Our study, although we have not followed the fracture rate, agrees in reporting lower TBS values in patients with lower 25OHD levels, relating this to defects in bone quality.

Regarding BMD measurements, there were minor changes between the three studied groups followed for a mean average of 10 years. There was a slightly higher bone mass at the lumbar and femoral neck in the group with 25OHD insufficiency levels, and at mid-distal, 1/3 distal and total radius in the group with sufficiency levels, compared with the deficiency group. We have not found differences in serum 25OHD levels between smokers and non-smokers, neither between the number of cigarettes per day nor the studied parameters. We, therefore, cannot confirm that smoking seems to depress the serum levels of 25OHD as reported in healthy women aged 45–58 years [28]. Our results are in agreement with other reports in elderly subjects (mean age 70 years), in which smoking was unrelated to serum vitamin D [29]. The daily intake of calcium was similar between our three groups studied and is in the range of the mean calcium intake (629 ± 21 SE) reported for aged women in a recent national survey in Spain [30].

The possible deleterious effect over the bone of long-term LT4 therapy in patients with PTC and low levels of 25OHD is currently being debated. Our finding that 64% of patients showed TBS values in the degraded and partially degraded range, even without significant bone loss, enhances the necessity of an adequate follow-up of these patients with trabecular studies to evaluate the risk of fractures. To our knowledge, no previous reports have evaluated the effects of 25OHD on TBS according to the categorization of vitamin D deficiency, insufficiency, and normal levels in patients with thyroid cancer and long-term treatment with LT4. Our finding of a deleterious TBS in relation with lower 25OHD, deficit, and insufficiency, suggest a possible effect of vitamin D on the bone microarchitecture, that could contribute to the increased fracture risk.

There are some limitations to our study. We have performed a cross-sectional analysis, excluding several conditions that could affect bone metabolism. Serum 25OHD was only measured at a one-time point and fracture could not be analyzed. We have only addressed postmenopausal women with papillary thyroid cancer. A control group with postmenopausal women treated with LT4 and matched 25OHD with our cohort, could not be possible due to ethical reasons. The strength of this study is that it provides information not only about BMD but also on bone microarchitecture (TBS) in patients with long-term follow-up. Also, we have compared our results with the national reference data for BMD and TBS.

## 5. Conclusions

We have found a serum 25OHD level of deficiency in 43%, and sufficiency in 36% of postmenopausal women with total thyroidectomy and long-term LT4 therapy due to PTC. Deficiency and insufficiency of vitamin D were associated with abnormal trabecular microarchitecture compared with patients with 25OHD levels >30 ng/mL and independently of BMD values. Our data support 25OHD levels >30 ng/mL to maintain adequate bone microarchitecture. We suggest that in patients with low vitamin D levels and PTC, TBS should be evaluated along with BMD.

## Figures and Tables

**Figure 1 medsci-08-00049-f001:**
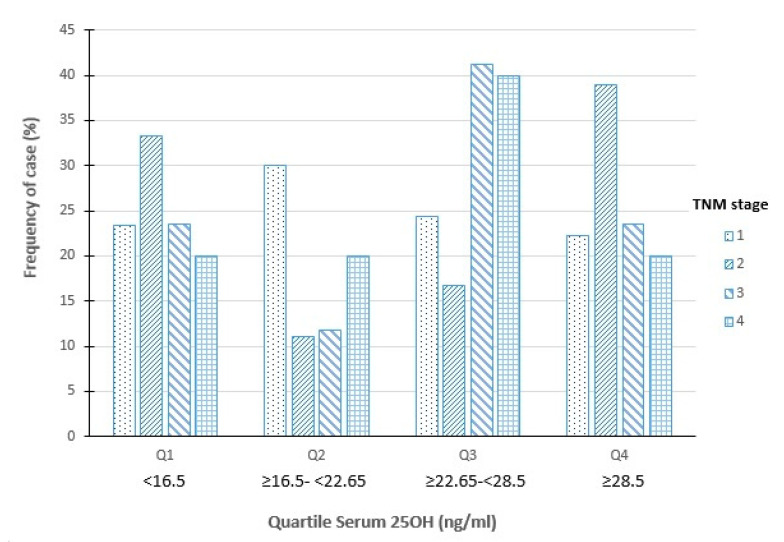
Frequency of cases distributed according to serum 25OHD quartiles and stages of thyroid cancer TNM (Fisher’s test *p* = 0.457).

**Table 1 medsci-08-00049-t001:** Baseline clinical and densitometric characteristics of papillary thyroid cancer patients.

Parameters	Data
N°	134
Age (years)	64.07 ± 10.8
Duration disease (years)	12.25 + 6.19
BMI (kg/m^2^)	28.62 + 5.5
Smoking status	
Never	118 (88.06%)
Current	16 (11.94%)
Diet calcium ingestion (mg/daily)	583.53 ± 294
Serum Calcium (mg/dL)	9.08 ± 0.4
Serum Phosphate (mg/dL)	3.35 ± 0.8
Serum Creatinine (mg/dL)	0.76 ± 0.2
Serum TSH (IU/mL)	0.92 ± 1.72
Serum fT4 (ng/dL)	1.64 + 0.25
Serum PTH (pg/mL)	51.14 ± 21.01
Serum Alkalione Phosphatase	105.75 ± 44.9
Serum BAP (U/L)	25.96 ± 11.9
Urinary calcium (mg/24 h)	157.22 ± 139.4
Serum 25OHD (ng/dL)	23.09 ± 7.97
LT4 dose/weight (µg/Kg per day)	1.68 + 0.42
*Bone parameters*	
L-BMD (g/cm^2^)	0.88 ± 0.13
FN-BMD (g/cm^2^)	0.70 ± 0.12
TH-BMD (g/cm^2^)	0.85 ± 0.13
UDR-BMD (g/cm^2^)	0.39 ± 0.06
MIDR-BMD (g/cm^2^)	0.51 ± 0.07
1/3 DR-BMD (g/cm^2^)	0.62 ± 0.07
TR-BMD (g/cm^2^)	0.49 ± 0.06
L-T-score	−1.45 ± 1.15
FN-T-score	−1.38 ± 1.04
TH-T-score	−0.71 ± 1.03
UDR-T-score	−0.87 ± 1.05
MIDR-T-score	−1.84 ± 1.28
1/3 DR-T-score	−1.24 ± 1.23
TR-T-score	−1.58 ± 1.20
TBS	1.27 ± 0.13

BAP, bone alkaline phosphatase. BMI, body mass index. BMD, bone mineral density. TBS, trabecular bone score. Serum fT4, free thyroxine. LT4, levothyroxine. TSH, thyrotrophin stimulating hormone. PTH, parathyroid hormone. 25OHD, 25-hydroxyvitamin D3. Mean values ± SD. BMI = body mass index. BMD = bone mineral density. TBS = trabecular bone score. Serum fT4 = free thyroxine. L-BMD = lumbar BMD; FN-BMD = femoral neck BMD; TSH-BMD = total hip BMD; 1/3 RD BMD = 1/3 radius BMD. L-T score = lumbar T score; FN T score = femoral neck T score; TH score = total hip T score.

**Table 2 medsci-08-00049-t002:** Clinical and densitometric parameters of thyroidectomized patients with papillary thyroid cancer according to the categorization of serum 25OHD levels.

Group Serum 25OHD Levels	<20 ng/mL (a)	20–29 ng/mL (b)	>30 ng/mL (c)
Anthropometric parameters			
N°	58	48	28
Age (years)	66.5 ± 10	60.6 ± 10.6 *	64.9 ± 11.4 ≠
Duration of disease (years)	13 ± 0.5	11.2 ± 5.7	12.4 ± 6.9
BMI (kg/m^2^)	29.6 ± 5.5	28.8 ± 5.9	26.3 ± 3.9
Smoking status			
Never	51 (87.93%)	42 (87.5%)	25 (89.29%)
Current	7 (12.07%)	5 (12.5%)	3 (10.7%)
Diet calcium (mg/daily)	535.95 ± 338.4	609.43 ± 262.9	626.25 ± 258.22
Serum calcium (mg/dL)	9.09 ± 0.44	9.01	9.2 ± 0.3
Serum phosphate (mg/dL)	3.31 ± 0.69	3.32 ± 0.9	3.5 ± 0.6
Serum creatinine (mg/dL)	0.78 ± 0.21	0.71 ± 0.20	0.77 ± 0.16
Serum TSH (µU/mL)	0.78 ± 0.21	0.88 ± 1.96	1.27 ± 1.8 ≠
Serum fT4 (ng/dL)	1.67 ± 0.24	1.68 ± 0.27	1.53 ± 0.2 ∆
Serum BAP (U/L)	29.50 ± 14.04	23.47 ± 8.87 ¥	22.14 ± 9.7 ≠£
Serum PTH (pg/mL)	57.65 ± 22.68	45.88 ± 19.8 *	47.13 ± 16.0 ∆
Urinary calcium (mg/dL)	129.36 ± 94.16	159.21 ± 92.21	220.3 ± 250
Serum 25OHD (ng/mL)	15.64 ± 2.98	25.27 ± 2.69	34.78 ± 3.62
LT4 dose/weight (µg/kg/d)	1.68 ± 0.38	1.74 ± 0.34	1.59 ± 0.61
L-BMD (g/cm^2^)	0.89 ± 0.14	0.91 ± 0.11 ¥	0.83 ± 0.11 ∆
FN-BMD (g/cm^2^)	0.70 ± 0.11	0.72 ± 0.12 ¥	0.64 ± 0.10 ∆
TH-BMD (g/cm^2^)	0.86 ± 0.14	0.88 ± 0.13 *	0.78 ± 0.11 ∆
UDR-BMD (g/cm^2^)	0.39 ± 0.07	0.41 ± 0.06	0.37 ± 0.04 ≠
MIDR-BMD (g/cm^2^)	0.49 ± 0.07	0.53 ± 0.06 ¥	0.49 ± 0.07 ≠£
1/3 DR-BMD (g/cm^2^)	0.60 ± 0.07	0.65 ± 0.07	0.60 ± 0.08 ≠€
TR-BMD (g/cm^2^)	0.48 ± 0.07	0.52 ± 0.06	0.48 ± 0.06 £
L-T score	−1.33 ± 1.26	−1.28 ± 1.03	−1.97 ± 1.00 ∆£
FN-T score	−1.34 ± 0.98	−1.17 ± 1.11	−1.85 ± 0.90 ∆£
TH-T score	−0.62 ± 0.98	−0.49 ± 1.04	−1.32 ± 0.89 ∆£
UDR-T score	−0.95 ± 1.18	−0.59 ± 0.97	−1.20 ± 0.75 ≠
MIDR-T score	−2.14 ± 1.30	−1.35 ± 1.16 ¥	−2.07 ± 1.22
1/3 DR-T score	−1.51 ± 1.19	−0.78 ± 1.15 ¥	−1.52 ± 1.28 ≠
TR-T score	−1.82 ± 1.22	−1.15 ± 1.11 ¥	−1.86 ± 1.12
TBS	1.24 ± 0.13	1.27 ± 0.13 ¥	1.31 ± 0.11 £

* a/b < 0.01; ¥ a/b < 0.05; ∆ b/c ≤ 0.01; ≠ b/c < 0.05; € a/c < 0.01; £ a/c < 0.05; BAP, bone alkaline phosphatase. BMI, body mass index. BMD, bone mineral density. TBS, trabecular bone score. Serum fT4, free thyroxine. LT4, levothyroxine. TSH, thyrotrophin stimulating hormone. PTH, parathyroid hormone. 25OHD, 25-hydroxyvitamin D3. Mean values ± SD. BMI = body mass index. BMD = bone mineral density. TBS = trabecular bone score. Serum fT4 = free thyroxine. L-BMD = lumbar BMD; FN-BMD = femoral neck BMD; TH-BMD = total hip BMD; 1/3 RD BMD = 1/3 radius BMD. L-T score = lumbar T score; FN T score = femoral neck T score; TH score = total hip T score.

**Table 3 medsci-08-00049-t003:** Distribution of papillary thyroid cancer patients, with osteoporosis, osteopenia, and normal bone mineral density (BMD), according to the levels of 25OHD.

Serum 25 OHD ng/mL	Osteoporosis	Osteopenia	Normal
<20 ng/mL	3 (2.2%)	47 (35%)	8 (6%)
20–29 ng/mL	0	39 (29%)	9 (6.7%)
>30 ng/mL	1 (0.8%)	18 (13.4)	9 (6.7%)
Total	4 (3%)	104 (77.6%)	26 (19.4%)

Number of patients and percent (%) in each group in respect of the total of studied patients.

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
