# Peer review of "Association of Low Serum 25OHD Levels with Abnormal Bone Microarchitecture in Well-Differentiated Thyroid Cancer"

_medsci, 2020, doi:10.3390/medsci8040049_

Round 1
Reviewer 1 Report
In the paper " Association of low serum 25OHD levels with 3 abnormal bone microarchitecture in well 4 Differentiated Thyroid Cancer” Hawkins-Carranza et al., provide for the first time the association between deleterious trabecular bone status and low levels of 25OHD. Overall the manuscript is well written but please take care to check over the all manuscript as there are several minor errors (extra/missing spaces, font type and size). The results are interesting, but the main problem of this study is the lack of healthy controls. I am aware that find controls is not easy, but I strongly suggest authors to select almost the same number of postmenopausal Caucasian subjects with some clinic features of patients but PTC free. Other minor comments are:
- Authors studied PTC not DTC. Please change accordly.
- Authors have to write the long name and no acronym for PTH and BAP.
Author Response
Reviewer 1
1) Overall the manuscript is well written but please take care to check over the all manuscript as there are several minor errors (extra/missing spaces, font type, and size)
This has been carefully done to accomplish this requirement.
2) The results are interesting, but the main problem of this study is the lack of healthy controls. I am aware that find controls are not easy.
We agree with Reviewer 1 recommendation. Unfortunately, our Ethical Committee did not allow us to obtain blood for this purpose from postmenopausal healthy women. Now in the present Covid-19 pandemic, this is even more difficult to make. The purpose of our research was to observe if there was a difference in Trabecular Bone Score and Bone Mineral Density, between the three different groups of patients with serum 25OHD levels categorized according to Scientific Societies. Mean levels of serum 25OHD in patients with Thyroid Cancer were lower than the reference values of our laboratory, and of that of the healthy population studied in a national survey (Ola et al. Nutrients 2017)
3) Authors studied PTC, not DTC. Please change accord. The author has to write the long name and no acronym for PTH and BAP
Although Papillary Thyroid Carcinoma, represents the major Differentiated Thyroid Cancer, we agree with Reviewer 1, and now we have changed this in the manuscript, as well regarding the indications for PTH (Parathyroid Hormone) and BAP (Bone alkaline phosphatase).(Abstract line 22 and others)
Reviewer 2 Report
In this study the authors tried to evaluate the impact of low levels of vitamin D on the TBS in patients treated for DTC. They found low TBS in patients with DTC and long-term follow-up associated with low serum 25OHD levels and conclude that these low values should be restored for the prevention of fractures.
The paper faced with an interesting argument, however there are several issues that needs to be re-evaluated, limiting the publication of the study in its current form. Several limitations were present in order to demonstrate the proposed aim of this study. Too few patients were involved, all post-menopausal women without a control group and no longitudinal clinical, biochemical and urinary data were available. This makes difficult the critical review of the study.
Major comments
Line 132-141 and Tab.1 – In the statistical analysis section, it was specified that mean or median were used as appropriate and that normal distribution of the data were evaluated by KS test. This is correct, but data were always expressed as mead and SD and no median and IQR have been shown. Moreover, several data in several series are often non-parametric data, such as age, biochemical or urinary parameters. Do the authors confirms that all analyzed parameters have a normal distribution?
Line 132 and Tab 1 – Please specify how many 131-I treatment were performed and if in hypothyroidism or after rhTSH. These data could be very interesting because of the potential impact of these further therapies and in particular of the hypothyroidism on the bone microarchitecture.
Line 133 and Table 2 – The authors specify that all patients performed an L-T4 therapy to suppress TSH. However in the methods section and in the table 2 they specified that started with suppressive therapy but continued with a mild suppressive – substitutive therapy thereafter. Please clarify.
Line 133 - How many evaluation (mean) the authors performed of a single patient during the follow up? Did the measurement of TSH obtained at every evaluation? How many time TSH was evaluated in a single patient? Please clarify
Table 2 – Please correctly align the data of the table.
Table 2 – Patients with vitamin D deficiency were significantly older than patient with vitamin D insufficiency. This could led to a different prevalence of bone alterations between the two groups.
Line 269 – This is an assumption not true for this paper because of the TSH is not suppressed in the population study.
Minor comments
Some typos (line 116 – Kruskal-Wallis, line 139, etc)
Author Response
Reviewer 2
- Several limitations were present in order to demonstrate the proposed aim of this
study. Too few patients were involved, all postmenopausal women without a control group, and no longitudinal clinical, biochemical and urinary data were available.
In this cross-sectional study we have voluntarily enrolled females patients that are more prevalent in thyroid cancer, and with the postmenopausal stage due to several reasons. It has been shown that bone loss is not found in males and premenopausal women with thyroid cancer and Levo-thyroid treatment (Sugitaniet al, Surgery 2011, Reverter et al. Endocr.Relat Cancer 2005). Also, in some healthy women, menopause induces a reduction of bone density, concerning estrogen deprivation which induces the release of cytokines (TNF-α and interleukins1 and7) which promotes osteoclastogenesis together with osteocyte apoptosis(Khalid et al, Bone2016). Therefore for our study purpose, postmenopausal women were a sensible remodeling target. Finally, we agree with Reviewer 2 of the great value of performing a future longitudinal study.
- Line132-141and tab.1.In the statistical analysis section, it was specified that mean
or median were used as appropriate and that normal distribution of the data was evaluated by the KS test. This is correct, but data were always expressed as mean and SD and no median and IQR have been shown. Moreover, several data in several series are often non-parametric data, such as age, biochemical or urinary parameters. Do the authors confirms that all analyzed parameter have a normal distribution?
There was a normal distribution in parameters. Parameters without a normal distribution as time of walking, calcium ingestion, urinary calcium, and alkaline phosphatase did not follow a normal distribution. This variable is therefore expressed as median and interquartile range.
3)Line 32 and Tab 1. Please specify how many131 treatment were performed and if in hypothyroidism or after rhTSH.This data could be interesting because of the potential impact of thse further therapies and in particular of the hypothyroidismon the bone microarchitecture.
All patients followed in our center for Differentiated Thyroid Cancer, follow a protocol for the study of possible recurrences, using rhTSH. We agree with Reviewer 2, about the possible effect of hypothyroidism on the bone. In our study patients with diseases that affect bone (hypothyroidism) were excluded. Recently a review has emphasized that hypothyroidism has been shown to decrease BMD in the majority of studies (Apostu D, Diagnostics 2 020). On the other hand, hypothyroidism leads to a reduction in bone remodeling, as evidenced by dynamic histomorphometric studies of iliac crest bone biopsies, as well as measures of bone turnover markers. By HRpQCT, the trabecular number is greater, and trabecular separation is lower, which agrees with the finding of increased TBS in hypoparathyroidism (Barbara C Silva 1 2 3, John P Bilezikian, Rev Endocr Metab Disord 2020 Nov 16). This is one of the reasons for selecting the rhTSH test in these patients.
- The authors specify that all patients performed an LT4 therapy to suppress TSH.
Following the suggestions of Reviewer 2, we have now clarified that patients had a moderate suppression therapy at the cross-sectional study, due to the early stage of diagnosis of cancer (line 274),
20 patients had 131-I treatment: 7 in the group with serum 25OHD <20 ng/ml, 6 in the group with >30and another 6 with serum levels >30 ng/ml without intergroup significant differences in total doses (lines 132-140)
- Please correctly align the data in table 2.
We have resubmitted the table, that was initially correctly aligned.
- Patients with vitamin D deficiency were significantly older than patient with
vitamin D insufficiency.
Patients with sufficiency 25OHD levels (>30) were significantly older than patients with insufficiency levels (20-29) and not different from patients with deficiency. Also, all data were age-adjusted.
- Line 260. This is an assumption not true for this paper because of the TSH is
not suppressed in the population study.
We agree with Reviewer 2, and we have now written: long-term L-T4 therapy in patients with PTC(line 274).
Minor comments.
Some typos (line 116)- Kruskal Wallis, line 139, etc.
We apologize for these typing mistakes that are now corrected.
Reviewer 3 Report
The study by Federico Hawkins-Carranza et. al. entitled "Association of low serum 25OHD levels with abnormal bone microarchitecture in well Differentiated Thyroid Cancer" is an interesting and well written work describing the influence of D3 vitamin insufficiency/defficiency on bone microarchitecrure in DTC. The results seem to confirm previous findings from literature. I haven't found any mistakes in the work, apart from some minor spelling, punctuation mistakes and I think that the work is suitable for publication in Medical Sciences.
Author Response
Reviewer 3.
The study by F Hawkins et al. is an interesting and well-written work describing the influence of vitamin D3 insufficiency/deficiency on bone microarchitecture in DTC. The results seem to confirm previous findings from the literature. I haven’t found any mistakes in the work, apart from some minor spelling, punctuation mistakes and I think that the work is suitable for publication in Medical Sciences.
We have made an effort to correct spelling and punctuation mistakes, with the collaboration of the Medical Professional Writer.
We want to thank the Assistant Editor and the Reviewers, for the time and consideration of our manuscript.
Sincerely
Federico Hawkins Carranza, MD, PhD
Chairman Medicine, Complutense University Madrid, Spain
Research Institute i+12, University Hospital 12 de Octubre, Madrid.
Round 2
Reviewer 1 Report
Thanks to tha authors to address my cmmoments.
Reviewer 2 Report
I thank to the authors for partially improving their manuscript according to the reviewers' suggestions.
Some data remains unclear and for the initial design of the study, this cannot be improved. However, I appreciate the extensive responses by the authors.